# Influence of the Alcohols on the ZnO Synthesis and Its Properties: The Photocatalytic and Antimicrobial Activities

**DOI:** 10.3390/pharmaceutics14122842

**Published:** 2022-12-18

**Authors:** Ludmila Motelica, Bogdan-Stefan Vasile, Anton Ficai, Adrian-Vasile Surdu, Denisa Ficai, Ovidiu-Cristian Oprea, Ecaterina Andronescu, Dan Corneliu Jinga, Alina Maria Holban

**Affiliations:** 1National Research Center for Micro and Nanomaterials, University Politehnica of Bucharest, 060042 Bucharest, Romania; 2National Research Center for Food Safety, University Politehnica of Bucharest, Splaiul Independentei 313, 060042 Bucharest, Romania; 3Faculty of Chemical Engineering and Biotechnologies, University Politehnica of Bucharest, 1-7 Polizu St., 011061 Bucharest, Romania; 4Academy of Romanian Scientists, Ilfov Street 3, 050044 Bucharest, Romania; 5Department of Medical Oncology, Neolife Medical Center, Ficusului Bd. 40, 077190 Bucharest, Romania; 6Microbiology and Immunology Department, Faculty of Biology, University of Bucharest, 077206 Bucharest, Romania

**Keywords:** ZnO, solvothermal, alcohol, photocatalysis, antimicrobial, methylene blue, methyl orange, eosin Y, crystal violet, rhodamine B

## Abstract

Zinc oxide (ZnO) nanomaterials are used in various health-related applications, from antimicrobial textiles to wound dressing composites and from sunscreens to antimicrobial packaging. Purity, surface defects, size, and morphology of the nanoparticles are the main factors that influence the antimicrobial properties. In this study, we are comparing the properties of the ZnO nanoparticles obtained by solvolysis using a series of alcohols: primary from methanol to 1-hexanol, secondary (2-propanol and 2-butanol), and tertiary (tert-butanol). While the synthesis of ZnO nanoparticles is successfully accomplished in all primary alcohols, the use of secondary or tertiary alcohols does not lead to ZnO as final product, underlining the importance of the used solvent. The shape of the obtained nanoparticles depends on the alcohol used, from quasi-spherical to rods, and consequently, different properties are reported, including photocatalytic and antimicrobial activities. In the photocatalytic study, the ZnO obtained in 1-butanol exhibited the best performance against methylene blue (MB) dye solution, attaining a degradation efficiency of 98.24%. The comparative study among a series of usual model dyes revealed that triarylmethane dyes are less susceptible to photo-degradation. The obtained ZnO nanoparticles present a strong antimicrobial activity on a broad range of microorganisms (bacterial and fungal strains), the size and shape being the important factors. This permits further tailoring for use in medical applications.

## 1. Introduction

Zinc oxide (ZnO) is among the most studied materials, representing a workhorse in the domain of nanomaterials [1,2]. At the same time, the FDA (Food and Drug Administration, Silver Spring, MD, USA) classifies ZnO as GRAS (generally recognized as safe). Therefore, ZnO has become one of the preferred choices for many applications [3,4,5]. The properties of ZnO are generating applications in many different domains, from the food industry to health care, and this is the main cause for the high level of interest it enjoys [6,7,8,9]. The theoretical band-gap of 3.37 eV places ZnO’s absorption in the (ultraviolet) UV domain [10]. The high absorbance, between 200–400 nm, makes it a natural choice for sunscreen and protection treatments, alongside TiO_2_ [11,12,13]. As a semiconductor, due to its transparent nature, ZnO is used in optoelectronic devices [14,15]. Its white color and chemical stability support its use as a pigment in various paints and anticorrosion coatings [6].

The literature presents many reports on various synthesis methods: precipitation [11], thermal decomposition [16,17], chemical vapor deposition, spray pyrolysis [18], natural extracts, ionic liquids [19], forced hydrolysis or solvolysis [10], sol-gel, or microwave [20]. Each method has advantages and disadvantages [21]. For example, in the precipitation method, a high yield of ZnO can be obtained, but the samples contain impurities such as alkaline ions, and calcination is required to transform all Zn(OH)_2_ [22,23,24,25]. By using natural extracts, the batch is usual small, and some residual organics remain trapped on the nanoparticles’ surface. In spray pyrolysis, although the crystallite size is under 100 nm [26,27,28,29], the size of the particles exceeds the nano domain.

Forced solvolysis in alcohols without NaOH addition is seldom reported, and studies still need to be carried out. In one such study Tonto P. et al. used primary alcohols with 4, 6, 8, and 10 carbon atoms in an autoclave at high temperatures of 250–300 °C for 2 h to obtain nanorods of ZnO [30]. The X-ray diffraction (XRD) analysis indicates that at lower temperatures, the final product contains zinc acetate impurities. Even at high temperatures, the XRD for the sample obtained in n-butanol presents a large split of the peaks. In a more recent study, Saric A. et al. used primary alcohols with 2, 3, 4, 5, and 8 carbon atoms, also in an autoclave at 170 °C for 4 h, to obtain spherical and rod-like nanoparticles [31]. The theoretical modeling of the process indicated 1-butanol as the solvent in which the smallest crystallite is obtained. Nevertheless, none of these studies were followed up by investigation of the photocatalytic or antibacterial activities.

The presence of various crystalline defects introduces additional electronic levels between conduction band (CB) and valence band (VB), making ZnO nanoparticles an effective photocatalyst under UV light, like TiO_2_ [32,33,34], but more important under visible light, which suggests its use for water purification systems or self-cleaning surfaces [3,11,35,36]. Strong photocatalytic activity is related to the generation of reactive oxygen species (ROS). Many materials generate ROS in the presence of light [37], and ZnO is one with a high capacity of ROS generation. Such materials usually exhibit strong antimicrobial activity. The exact mechanism of antimicrobial activity for ZnO is not well established, but at least two separate pathways exist [38]. Besides the obvious contribution of ROS, there are numerous reports about ZnO’s intrinsic antimicrobial activity under dark conditions, most probably due to mechanical puncture of the cellular membrane, internalization, and interference with essential constituents of the microbial cell [39,40,41,42,43]. In presence of light, the ZnO nanoparticles will produce ROS that are responsible for the oxidative stress induced to the microorganisms, which is damaging to the cellular membrane. The antibacterial activity is well studied, but the antimycotic activity is rarely investigated [41]. Due to its potent antimicrobial activity, ZnO is studied as a constituent in topical ointments [44,45], wound dressing [46,47,48,49,50], drug delivery systems [51,52,53,54], but also as a component in antimicrobial packaging [55,56,57,58,59,60,61] or antimicrobial textiles [62,63,64,65], as well as in dentistry [66,67,68,69].

In this study, we employed a force solvolysis method, starting from zinc acetate as a precursor and various alcohols (primary alcohols from methanol to 1-hexanol, secondary, and tertiary alcohols), without an autoclave. The optimization of parameters accounted for reflux time and rest time. The obtained ZnO samples were characterized by thermogravimetric analysis (TG-DSC), X-ray diffraction (XRD), transmission electron microscopy (TEM), ultraviolet-visible spectroscopy (UV-Vis), Fourier transform infrared spectroscopy (FTIR), and photoluminescence (PL), and the samples were tested for photocatalytic and antimicrobial activities. The sample with the best photocatalytic activity against methylene blue was used further for a series of common pollutant dyes (methyl orange, eosin Y, rhodamine B and gentian (crystal) violet) to obtain a direct comparison among them, which would facilitate cross-comparisons among other photocatalytic reports in the literature. The antimicrobial activity was determined against Gram-negative strains *Escherichia coli*, *Salmonella enterica* serovar Typhimurium, and *Pseudomonas aeruginosa*, which are considered among the most common cause of food poisoning [70,71,72,73,74], Gram-positive strains *Staphylococcus aureus*, *Bacillus cereus, Bacillus subtilis*, and *Enterococcus faecalis*, which are responsible for many implants’ failure [75,76], skin infections [77], or foodborne illnesses [78], and a fungal strain *Candida albicans*, responsible for oral and vaginal candidiasis [79,80,81].

## 2. Materials and Methods

Zinc acetate dihydrate (Zn(CH_3_COO)_2_·2H_2_O) with 99.9% purity was purchased from Merck (Merck Group, Darmstadt, Germany). The alcohols, methanol (C1), ethanol (C2), 1-propanol (C3), 2-propanol (sC3), 1-butanol (C4), 2-butanol (sC4), tert-butanol (tC4), 1-pentanol (C5), and 1-hexanol (C6), were used as received from Sigma (Redox Lab Supplies Com SRL, Bucharest, Romania), without further purification. Phosphate-buffered saline (PBS), nutrient broth, and agar were obtained from Sigma Aldrich (Redox Lab Supplies, Bucharest, Romania).

For the ZnO synthesis, 5.000 g of Zn(CH_3_COO)_2_·2H_2_O were mixed with 50 mL alcohol (for each type in a separated flat bottom flask) and were kept under magnetic stirring at 60 °C. After 24 h, the flask was removed from the heater and allowed to rest another 24 h at room temperature. The obtained precipitate was centrifuged and washed with absolute ethanol thrice, followed by drying in an electrical oven at 80 °C. The obtained ZnO nanopowders were labeled as a function of alcohol used, as in Table 1:

The syntheses made in secondary and tertiary alcohols were considered unsuccessful, and based on the analysis results, they were further discarded from the study.

An STA 449C Jupiter equipment from Netzsch (Selb, Germany) was used for the thermogravimetric analysis coupled with differential scanning calorimetry (TG/DSC). The samples (~20 mg of dry powder) were placed in an open Al_2_O_3_ crucible and heated up at a 10 °C∙min^−1^ rate until 900 °C, in dynamic atmosphere (flow of 50 mL∙min^−1^ of dried air—20% O_2_ and 80% N_2_). An empty alumina crucible was used as a reference.

A Nicolet iS50R spectrometer (Thermo Fisher Scientific, Waltham, MA, USA) was used to record the Fourier transform infrared spectra (FTIR). All measurements were performed at room temperature using the attenuated total reflection (ATR) accessory (Thermo Fisher Scientific, Waltham, MA, USA). Each spectrum is the average of 32 sample scans between 400 and 4000 cm^−1^_,_ with a resolution of 4 cm^−1^.

A JASCO (Easton, PA, USA) V560 spectrophotometer equipped with a 60 mm integrating sphere (ISV-469) was used to record the diffuse reflectance spectra (UV-Vis). All the measurements were made in the domain 200–900 nm, with a speed of 200 nm min^−1^.

A Perkin Elmer LS55 (Perkin Elmer, Waltham, MA, USA) fluorimeter was used to record the photoluminescence spectra (PL). A Xe lamp was used as an excitation source at ambient temperature. The excitation wavelength was 320 nm. The emission spectra were recorded in the domain 350–700 nm, with a scan speed of 200 nm min^−1^ and a 350 nm cut-off filter.

Information about the crystalline phases were obtained with a PANalytical Empyrean equipment (from Malvern PANalytical, Bruno, Nederland) using a Bragg–Brentano geometry, equipped with a Cu anode (λ_CuKα_ = 1.54184 Å) X-ray tube and hybrid monochromator (Ge220). The X-ray diffractograms (XRD) were recorded in the 2θ range 10–70°, with a step of 0.02° and the time per step of 0.1 s.

A Tecnai G2F30 S-TWIN high-resolution transmission electron microscope (TEM) from FEI (FEI Company, Eindhoven, The Netherlands) was used to acquire the images for ZnO nanoparticles. The equipment was operated with a Schottky field emitter at 300 kV acceleration voltage. The microscope has a point resolution of 2Å and line resolution of 1.02 Å.

The photocatalytic activity was determined against methylene blue (MB) 3.125 × 10^−5^ M (10 mg/L), methyl orange (MO) 6.110 × 10^−5^ M (20 mg/L), eosin Y (EY) 1.235 × 10^−5^ M (8 mg/L), gentian violet (GV) 1.222 × 10^−5^ M (5 mg/L) and rhodamine B (RhB) 2.505 × 10^−5^ M (12 mg/L) solutions. A volume of 10 mL solution of organic dye was added over each ZnO sample (weighting 20 mg). The adsorption–desorption equilibrium was reached after 30 min under magnetic stirring in dark conditions. The irradiation process was made with a fluorescent lamp of 160 W/2900 lm, produced by LOHUIS^®^ (Lohuis, Bucharest, Romania), with a color rendering index of >60 and temperature of 3200 K, situated at a distance of 10 cm. At certain time intervals, a volume of solution was removed and placed in a 10 mm quartz cuvette. A JASCO (Easton, PA, USA) V560 spectrophotometer was used to record the spectrum of each sample, with a speed of 200 nm∙min^−1^.

Assessment of antimicrobial activity was performed using eight reference microorganisms by American Type Culture Collection (ATCC, Manassas, VA, USA): one fungal strain, *Candida albicans* (*C. albicans*) ATCC 10231; three Gram-negative bacterial strains of *Escherichia coli* (*E. coli*) ATCC 25922; *Salmonella enterica* serovar Typhimurium (*S. typhimurium*) ATCC 14028; *Pseudomonas aeruginosa* (*P. aeruginosa*) ATCC 27853; four Gram-positive bacterial strains of *Staphylococcus aureus* (*S. aureus*) ATCC 25923; *Bacillus cereus* (*B. cereus*) ATCC 19659; *Bacillus subtilis* (*B. subtilis*) ATCC 6633; and *Enterococcus faecalis* (*E. faecalis*) ATCC 29212. In order to determine the antimicrobial activity of ZnO nanoparticles, we employed an adapted diffusion assay, presented in [82,83], respecting the general rules exposed in the CLSI 2020.

Petri dishes containing nutritive agar were swab inoculated with a standardized inoculum, prepared as bacterial suspensions of 1.5 × 10^8^ CFU/mL (0.5 McFarland) in sterile saline solution (0.9% NaCl). From a previously obtained ZnO suspension (10 mg/mL), drops with a volume of 10 μL were placed on the inoculated Petri dishes, and these were incubated for 24 h at 37 °C. The diameter of the zone of inhibition of growth developed around each drop was measured (in mm) after incubation. All antimicrobial experiments were performed as triplicates. GraphPad Prism 9 for Windows 64-bit, version 9.1.1 (GraphPad Software, San Diego, CA, USA) was used to analyze the results developed. The analysis of variance (ANOVA) was used to compare the results, and Tukey’s multiple comparisons test (with *p* < 0.05) was considered statistically significant.

## 3. Results and Discussion

The obtained nanopowders were investigated to determine the purity, composition, and morphology of the nanoparticles. The photocatalytic activity against MB, MO, EY, GV, and RhB and the antimicrobial activity were determined for each ZnO sample. Correlations between structure, photocatalytic activity, and antimicrobial activity were made.

### 3.1. Thermal Analysis (TG-DSC)

Thermal analysis (TG-DSC) results were used to evaluate the purity of the nanopowders as obtained from the synthesis.

As the reactants include only the zinc acetate and the corresponding alcohol, the expected products are ZnO nanoparticles and eventually Zn(OH)_2_ and unreacted zinc acetate, together with polymeric intermediate species like Zn_5_(OH)_8_(CH_3_OO)_2_ or Zn_5_(CO_3_)_2_(OH)_6_ [10,84,85,86]. For the samples obtained in the C1-C6 alcohols, the thermal analysis results are presented in Figure 1. Results for individual thermal analyses made for samples obtained in primary alcohols are given in Appendix A, and the principal data (mass loss and effect temperatures) are listed in Appendix A.

The samples ZnO_C3, ZnO_C4, and ZnO_C5 are the most stable, losing between 0.13% (ZnO_C4) and 0.36% (ZnO_C3) of the initial mass in the temperature interval RT-240 °C. By contrast, the sample ZnO_C1 loses 1.48% in the same interval. This mass loss can be attributed to the elimination of residual alcohol or water molecules from the nanoparticles’ surface. After 240 °C, the acetate impurities start to decompose, releasing acetic acid and acetone, this process being accompanied by an endothermic effect on the DSC curve in the interval of 260–290 °C [87]. After 300 °C, the remaining organic impurities will be oxidized, together with the evolved products from decompositions as indicated by the exothermic peaks on the DSC curve in the 330–355 °C interval. The exothermic peaks from this interval present a shoulder, which can be attributed to the oxidation of residual carbonaceous mass from previous incomplete oxidation processes [88]. In all samples, an endothermic effect after 400 °C is probably related to the elimination of surface –OH moieties, densification, and crystallization processes [88]. The residual mass values are presented in Table 2.

The observable tendency is that residual mass (represented by ZnO) increases from ZnO_C1 to ZnO_C4 samples and then decreases towards the ZnO_C6 sample.

In Figure 2 are presented the results obtained for the syntheses in secondary and tertiary alcohols. The lower residual mass indicates that the ZnO in this case is highly impure. The residual mass varies from 75.75% for the sample ZnO_sC4 to 44.32% for the sample ZnO_sC3. For the samples ZnO_sC3 and ZnO_tC4, the endothermic peaks from the DSC curve between 90–200 °C indicates the existence of a decomposition step, while for ZnO_sC4, the first mass loss occurs over 200 °C and is represented by an oxidation process. These results, coupled with the XRD analysis, led us to discard from the study the samples obtained in secondary and tertiary alcohols.

For the sample ZnO_C4, which had the best purity, we have studied the influence of the refluxing and rest time, with the aim to optimize the synthesis conditions. The results are presented in Figure 3, where the first time is the reflux one and the second time is the rest one at room temperature.

As seen from the results, both reflux and rest time influence the purity of the sample, up to 24 h. A further increase to 30 h leads to no noticeable differences.

### 3.2. X-ray Diffraction Analysis (XRD)

The crystalline phase composition was determined by XRD analysis, Figure 4.

While for the nanopowders obtained in the primary alcohols, C1-C6, the single phase identified is a wurtzite ZnO structure [JCPDS card no. 80-0075] (Figure 4a), in the case of secondary and tertiary alcohols, additional peaks are presented (Figure 4b). The Miller indices label each of the nine peaks for the samples obtained in primary alcohols. The broadening of the XRD peaks indicates the nano nature of the sample’s particles. The calculated crystallite size can be considered the dimension of a coherently diffracting domain and might be different when compared with particle size [86].

Microstrain (ε), crystallite size (D), and the lattice parameters were calculated by Rietveld refinement (Table 3). The results indicate a decrease in the ratio *c*/*a* from the ZnO_C1 to ZnO_C4 samples, followed by an increase up to ZnO_C6. The positional parameter, *u*, is a measure of each atom displacement when comparing to the next on the *c* axis. The general rule is that when the *c*/*a* ratio decreases, the positional parameter, *u*, increases, which leads to higher microstrain values (ε) [86].

The dislocation density (δ), which is defined as the length of dislocation lines in a unit volume, practically represents the number of defects in the sample, being calculated as 1/D^2^ (where D is the crystallite size). The highest δ value is obtained for the ZnO_C4 sample and the lowest value for the ZnO_C6 sample. As the surface defects represent active centers in the photocatalytic and antimicrobial activities, the samples with the highest δ values are expected to be the most active.

In a previous theoretical study [31], the authors have indicated that the smallest nanoparticles will be obtained in 1-butanol (when using primary alcohols with two to five carbon atoms). In addition, the variation of the calculated nanoparticles’ size is identical to the one we report from our syntheses (D_C2_ > D_C3_ > D_C5_ > D_C4_).

### 3.3. Transmission Electron Microscopy (TEM)

In order to investigate the dimension and shape of the nanoparticles, TEM bright-field images were obtained for each sample (Figure 5). The particles’ size is in good correlation with the crystallite size value determined from XRD, which indicates that in fact, each nanoparticle is monocrystalline, with similar reports being found in literature [31]. The ZnO_C1 nanoparticles have a rounded shape and present a low tendency to agglomerate (Figure 5a).

By contrast, the ZnO_C2 nanoparticles present a higher tendency to form flower-like agglomerates (Figure 5b). The size of the nanoparticles decreases towards ZnO-C3 and ZnO_C4 samples (Figure 5c,d). The nanoparticles for ZnO-C3 and ZnO_C4 samples are mainly polyhedral, with agglomeration tendencies. By contrast, the ZnO_C5 sample presents nanoparticles with irregular shapes (Figure 5e) that evolve towards nano-rods in the ZnO_C6 sample (Figure 5f). In addition, the nanoparticles do not agglomerate for the C5 and C6 alcohols.

The differences among the nanoparticles’ shape are probably induced by the presence of aliphatic chains on the nanoparticles’ surface during the crystal growth phase. The longer aliphatic chains block the increase of the grains in certain directions, allowing the development of rod-shaped nanoparticles. A similar hypothesis was made in [30], where authors obtained even micron-sized rods in 1-decanol.

### 3.4. Spectroscopic Studies

#### 3.4.1. FTIR Spectroscopy

In order to evaluate the purity of the ZnO nanopowder, the FTIR spectra were recorded (Figure 6). The strong absorption peak under 500 cm^−1^ (457 and 410 cm^−1^) corresponds to the stretching vibrations of the Zn–O bond [11]. The weak peaks from 614 and 685 cm^−1^ are due to Zn-OH bending vibrations [89]. The weak peaks from 1435 and 1568 cm^−1^ are characteristic of the carboxylate group, symmetric ν_s_(COO) and asymmetric ν_as_(COO) vibrations, respectively. When the Δν between these two bands is in the 100–200 cm^−1^ interval, the carboxylate is in ionic form. This implies the presence of some acetate ions adsorbed on the nanoparticles surface, in concordance with the oxidative step and exothermic effect seen in TG-DSC analysis. The peak from 1340 cm^−1^ can be attributed to the symmetric bending (δ_s_CH_3_) vibration [85].

The weak, broad band from 3400 cm^−1^ is attributed to the presence of –OH moieties on the nanoparticles’ surface (stretching vibration υOH). The peak from 1033 cm^−1^ can be assigned to the -C-O stretching vibration, while the band from 900 cm^−1^ is due to the presence of tetrahedral Zn^2+^ ions [90].

#### 3.4.2. UV-Vis Spectroscopy

The ZnO nanopowders are white, and therefore present little absorption in the visible spectrum, but they exhibit a strong absorption band in the UV domain, with the peak at 366 nm (Figure 7a). This strong absorption under 400 nm is the reason why ZnO nanoparticles are used as key component of sunscreen cosmetics [91,92,93], but also in the textile industry [94,95,96] or as a protective coating [97]. This absorption band is related to the electron jump from valence band to conduction band and can be used to estimate the band-gap values for ZnO nanopowders.

By using the Kubelka–Munk function F (R), with the value calculated by Equation (1) where R is the diffuse reflectance of the sample:F(R) = (1 − R)^2^/2R(1)
we can determine the direct band-gap energy values by graphical extrapolation to [F(R)∙hν]^2^ = 0 (Figure 7b).

The determined band-gap energy values are smaller than the theoretical value of 3.37 eV, but still close to it [21,34,98]. The multiple surface defects of the nanoparticles, as indicated by the XRD study, induce additional electronic levels inside the band-gap, leading to decreased energy values. The literature usually reports similar lower values [99,100].

The calculated band-gap energy values (Table 4) indicate a similar trend to the one observed for the *c*/*a* ratio from Table 3: a decrease in the value from the ZnO_C1 to the ZnO_C4 sample, followed by an increase up to ZnO_C6. This permits us to hypothesize that the defect density has a direct influence on the band-gap energy values. Nanoparticles with lower values for the band-gap energy are expected to be better photocatalysts.

#### 3.4.3. Photoluminescence Spectroscopy

The photoluminescence (fluorescence) spectra of the ZnO nanopowders, obtained on excitation with 320 nm, are presented in Figure 8.

The ZnO nanoparticles present usually two emission zones, one in the visible domain and one in the ultraviolet region [101,102,103,104]. The UV emission is generally placed around 390–400 nm, the near-band-edge (NBE) location, and is generated by the free exciton recombination (recombination of the electron and hole pair that was previously generated by the absorption of a photon). The presence of various surface defects that can trap the free electrons and block the recombination are the cause of the decreased intensity of this band vs. the visible emission bands [105,106,107]. The visible emission of the ZnO nanoparticles, or deep-level emission (DLE), is assigned to the electronic levels generated inside the band-gap by the defects like oxygen vacancies (V_O_), zinc vacancies (V_Zn_), oxygen anti-sites (O_Zn_), oxygen interstitials (O_i_), or zinc interstitials (Zn_i_), the relative energetic position of them inside the band-gap being reported in literature [3,108]. Therefore, taking into account the reported energies and possible transition mechanisms from the literature, we made the following assignments: the V_Zn_ are responsible for the emission in the violet region, and transitions from Zn^+^_i_ to valence band (VB) generate the 2.71 eV (457 nm) band, while transitions from Zn^+^_i_ to V_Zn_ are responsible for the 2.57 eV (481 nm) emission.

The green emission from 2.42 eV (513 nm) and further is generated mainly by V_O_ and V_O+_ defects [3,105,107,109,110]. The strongest emission intensity is observed for the ZnO_C4 sample, indicating a higher density of surface defects.

The presence of surface defects can enhance the production of reactive oxygen species, especially from the oxygen vacancies, which leads to higher antimicrobial activities under light irradiation [111].

### 3.5. Photocatalytic Study

Under light irradiation, the ZnO nanoparticles produce reactive oxygen species (ROS) that degrade organic compounds and induce microbicidal activity against various microorganisms by breaking the cellular membrane and cytoplasm components [111]. Therefore, the photocatalytic capacity of ZnO nanoparticles is directly related to the antimicrobial activity and to the potential uses of ZnO in disinfection applications.

The photocatalytic activity of each sample was investigated against methylene blue solution (Figure 9). MB was chosen because it is a phenothiazine dye and has the tendency to form dimers. Therefore, the photocatalytic activity is investigated against both the monomer and the dimer, with their absorption maxima at 664 nm and 614 nm, respectively [112,113,114,115]. There are some literature reports that doped ZnO nanoparticles are capable to photo-decompose the monomer quicker than the dimer [14], the decomposition of the dimer being stronger only after the monomer is completely decomposed. This indicates a possible competition between monomer and dimer for the catalytic centers and a preference for the monomer form. In this research, both maxima are decreasing, so we can conclude that the monomer and the dimer are decomposed by the ZnO nanoparticles simultaneously, as depicted by the proposed mechanism (Figure 9h).

The photo-degradation reactions, for low-concentration solutions, exhibit an apparent first-order kinetics, ln (C_0_/C) = *k*_app_∙t, where C_0_ is the initial concentration of the dye, C is the dye concentration at time t (minutes), and *k*_app_ is the rate constant of apparent first-order reaction. The *k*_app_ values can thus be determined by plotting ln (C_0_/C) vs. time (Figure 10a). The *k*_app_-determined values and the coefficient of determination R^2^ values are given in Table 5. All ZnO samples exhibit strong photo-degradation activity, with a degradation efficiency between 94.97% for ZnO_C1 and 98.24% for ZnO_C4 (Table 5). The highest degradation efficiency was obtained for the ZnO_C4, as can be seen from Figure 9g and Table 5 values, most probably due to the highest density of surface defects. Such defects represent catalytic centers for the photo-degradation of the organic molecules. Comparing the results with literature reports, we can observe that high values of *k_app_* at 81∙10^−3^ min^−1^ [116], 58∙10^−3^ min^−1^ [114], or 39.26∙10^−3^ min^−1^ [117] are reported for the sample under UV irradiation but are still lower than the best value obtained in this study for ZnO nanoparticles obtained in 1-butanol or 1-pentanol, under visible light irradiation.

The photocatalytic reports in the literature are hard to compare, as there is no standard method related to the light used (UV or visible), light intensity, distance from light source, dye concentration, addition of H_2_O_2_ or photocatalyst quantity used. In order to obtain comparative information about the photocatalytic activity of the ZnO against various organic dyes (used as model pollutants), we have used the ZnO_C4 sample further against MO, EY, GV and RhB solutions (Figure 11).

The calculated parameters for each dye used are presented in Table 6. The best degradation efficiency was obtained for MB and MO solutions, while GV was the least susceptible to the photo-degradation. It is worth noting that all three compounds, EY, GV and RhB, are part of the same class of triarylmethane dyes, this common structure bestowing upon them a higher resistance to photo-degradation [118], although some literature reports indicate a high removal rate for titanium-doped activated carbon nanostructures against such dyes [119]. It is important to understand which organic dyes are not susceptible to the photo-degradation by the ZnO nanoparticles, as there are specific applications where such behavior is unwanted. For example, in the textile industry, ZnO nanoparticles can be used as a UV shield and as an antimicrobial agent when they are embedded in the fibrillary structure [120,121,122,123]. A photocatalytic activity in this case is not desirable, as the ZnO nanoparticles would degrade the organic support that they are meant to protect [124,125,126].

### 3.6. Antimicrobial Assay

The overuse of antibiotics led to microorganisms’ acquired resistance to them, which is one of the biggest problems of the 21st century. ZnO was indicated as a nano-antibiotic against photogenic bacterial and fungal strains, with a strong disinfection potential, even against antibiotic-resistant microorganisms [127]. As mentioned before, the ZnO nanoparticles’ antimicrobial mechanisms are based on ROS production under light irradiation, but are also dependent on size and morphology of the nanoparticles (larger ones being usually less active) [1,111]. The antimicrobial assay was made in the absence of light in this study; thus, we confirm that the antimicrobial activity of ZnO depends on the size and shape of the nanoparticles (Figure 12).

The smaller nanoparticles, like ZnO_C4, are more potent, as expected, when compared with the larger ones. At the same time, the rod-shaped nanoparticles present the strongest antimicrobial activity, most probably due to easier penetration of the cellular membrane, which leads to mechanical damage like puncture or rupture [111]. X. Zhu et al. reported that small nanoparticles (under 10 nm), agglomerated or not, present a higher level of ROS production when compared with ZnO nanorods five times larger, which exhibited the smallest production. Interestingly, large spherical nanoparticles (40 nm) were found to produce ROS at an intermediate level [116].

The fungal strain *C. albicans* was the least susceptible among the microorganisms, with under 10 mm diameter of inhibition zone, with no clear differences between the smallest nanoparticles and the rod-shaped ones. From the bacterial strains, *P. aeruginosa* exhibited the smallest diameter of inhibition zone (still over 10 mm), but this strain is known for its natural resistance to antimicrobials and is one of the most difficult pathogens to manage in wound infections [128]. *P. aeruginosa*, like *E. coli* and *S. typhimurium*, was more susceptible to the ZnO_C4 sample, indicating that smaller nanoparticles are more potent against Gram-negative strains.

The *E. faecalis* presented the largest zone of inhibition, over 30 mm, which confirms this strain’s susceptibility to ZnO nanoparticles. For the rest of the Gram-positive strains, *S. aureus, B. cereus*, and *B. subtilis*, both factors, size and shape, seem to affect the magnitude of the inhibition zone. Nevertheless, for Gram-positive strains, the rod-shaped nanoparticles exhibited the strongest activity, indicating the role of morphology in this case.

The literature reports multiple pathways for the antimicrobial activity of ZnO, as indicated in Figure 13. Production of ROS can take place at the interface between nanoparticles and the cellular membrane, after nanoparticles’ adhesion to the cellular wall [129]. In the case of the large particles, clusters, and in general nanoparticles that do not come in contact with the bacterial membrane, the production of ROS remains the main bactericidal mechanism, as the oxidative stress generated by the accumulation of the ROS can still damage the plasma membrane. Of course, the smaller nanoparticles, with their larger surface, generally produce more ROS and therefore exhibit a larger antimicrobial activity [130]. As previously reported, smaller nanoparticles tend to cluster near the bacterial cell, and this leads to higher bactericidal activity, due to ROS being produced next to the cell wall. However nanoparticles can penetrate cell membranes and become internalized [131]. ROS production can be resumed inside the microbial cell with more devastating effects. Therefore, the capacity to penetrate the bacterial membrane promotes the nanoparticle to a higher efficiency state, where ROS are delivered inside the cell. Adding to this oxidative stress, the mechanical damage of the plasma membrane, by puncturing or rupture, leads to the leakage of cytoplasm components (Figure 13), such as nucleic acids (DNA and RNA), proteins and K^+^, followed by the quick death of the cell [129].

Lastly, once inside the microbial cell, the zinc ions can be released from the nanoparticles’ surface and contribute to cell death by cytotoxicity by binding essential components and nutrients of the cell [129]. Elucidation of the real antimicrobial mechanism requires further investigations, but the obtained results permit us to hypothesize that both size and morphology play important roles. In a recent study [116], authors found that 6 nm spherical ZnO nanoparticles are more potent than ZnO nanorods with a length of 55 nm against *E. coli* and *S. aureus.* In addition, the group reported that the ZnO nanorods seem to downregulate the expression of Ag43 surface protein, responsible for bacterial adherence and biofilm formation, which adds to the possible antimicrobial pathways of ZnO.

Among the studied strains, the Gram-positive bacteria were found to be more sensitive than the Gram-negative ones to the ZnO nanoparticles, regardless of shape and size, but overall, the antimicrobial activity can be considered strong across a wide spectrum of microorganisms.

By combining the native antimicrobial activity of the ZnO nanoparticles with the generation of ROS under light irradiation, an even higher microbicide effect can be obtained, and such nanoparticles can be tailored for specific applications.

## 4. Conclusions

In this study, we report the influence of alcohol type on ZnO synthesis by forced solvolysis. While the primary alcohols are suitable for obtaining ZnO nanoparticles, from spherical to rod-like shape, the syntheses in secondary and tertiary alcohols were not successful in yielding pure ZnO.

While the size of the nanoparticles was in the range 34–54 nm, with the lowest dimensions for the 1-butanol synthesis, the shape of the nanoparticles was found to be spherical for methanol, changing to polyhedral up to 1-butanol, and becoming rod-like for 1-hexanol. This permits further tuning of the morphology as desired.

The highest photocatalytic activity against methylene blue was found for the ZnO obtained in 1-butanol, with a photo-degradation efficiency of 98.24% after 40 min. The comparative study among a series of usual model dyes revealed that triarylmethane dyes are less susceptible to photo-degradation, especially gentian violet and rhodamine B, with an efficiency just above 50%.

The antimicrobial assay indicates that ZnO samples obtained in 1-butanol and 1-hexanol have the best activities. The sample first has the smallest nanoparticles, while the former has the rod-shaped particles. This indicates that the shape of the nanoparticles plays an important role in defining the antimicrobial activity for ZnO, most probably due to the mechanical-damage-induced stress generated by the puncture and rupture of the cellular membrane. Gram-negative strains proved to be more susceptible to the size of the ZnO nanoparticles, while the Gram-positive strains were susceptible to both smaller size and rod-shaped nanoparticles, with a higher influence from the morphology. Nevertheless, the obtained ZnO nanoparticles present a strong antimicrobial activity on a broad range of microorganisms, both bacterial and fungal strains, permitting further use in medical applications.

## Figures and Tables

**Figure 1 pharmaceutics-14-02842-f001:**
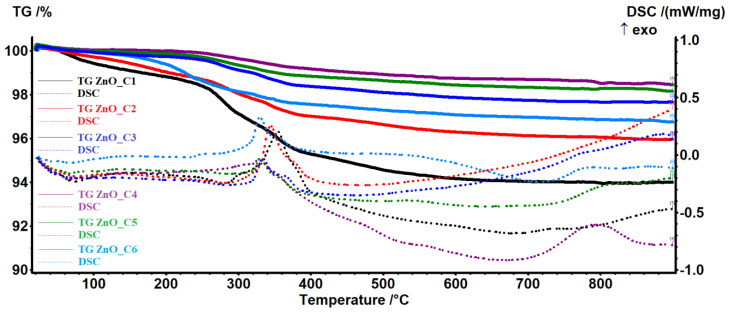
The thermal analysis, TG-DSC curves, for the nanopowders obtained in primary alcohols: ZnO_C1 in methanol; ZnO_C2 in ethanol; ZnO_C3 in 1-propanol; ZnO_C4 in 1-butanol; ZnO_C5 in 1-pentanol; ZnO_C6 in 1-hexanol.

**Figure 2 pharmaceutics-14-02842-f002:**
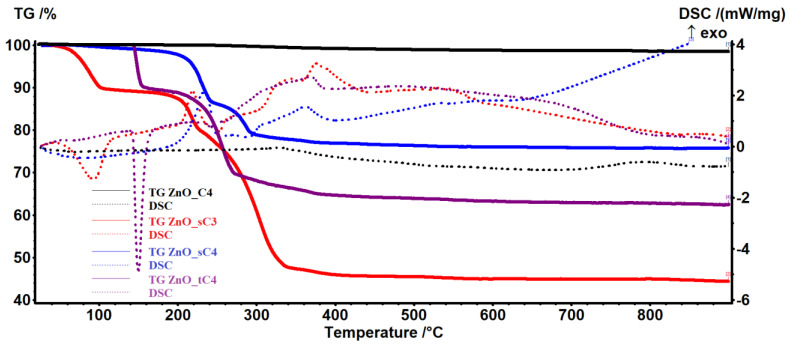
The thermal analysis, TG-DSC curves, for the nanopowders obtained in secondary and tertiary alcohols vs. 1-butanol: ZnO_sC3 in 2-propanol; ZnO_sC4 in 2-butanol; ZnO_tC4 in tert-butanol; and ZnO_C4 in 1-butanol.

**Figure 3 pharmaceutics-14-02842-f003:**
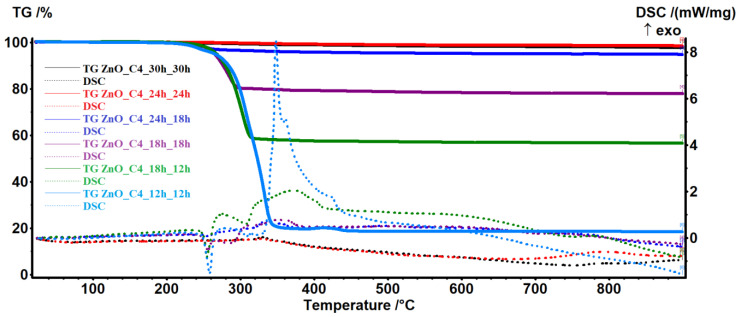
The thermal analysis, TG-DSC curves, for the ZnO_C4 nanopowders (ZnO obtained in 1-butanol) at different refluxing and rest times.

**Figure 4 pharmaceutics-14-02842-f004:**
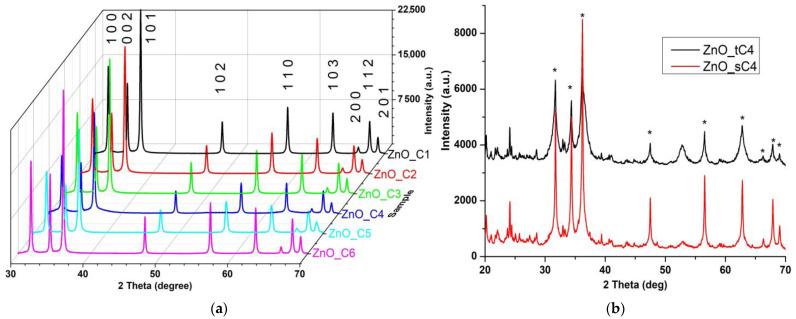
The XRD patterns for the nanopowders obtained in: (**a**) primary alcohols ZnO_C1 in methanol; ZnO_C2 in ethanol; ZnO_C3 in 1-propanol; ZnO_C4 in 1-butanol; ZnO_C5 in 1-pentanol; ZnO_C6 in 1-hexanol; (**b**) nanopowders in secondary and tertiary alcohols ZnO_sC4 in 2-butanol and ZnO_tC4 in tert-butanol (* denotes ZnO peaks).

**Figure 5 pharmaceutics-14-02842-f005:**
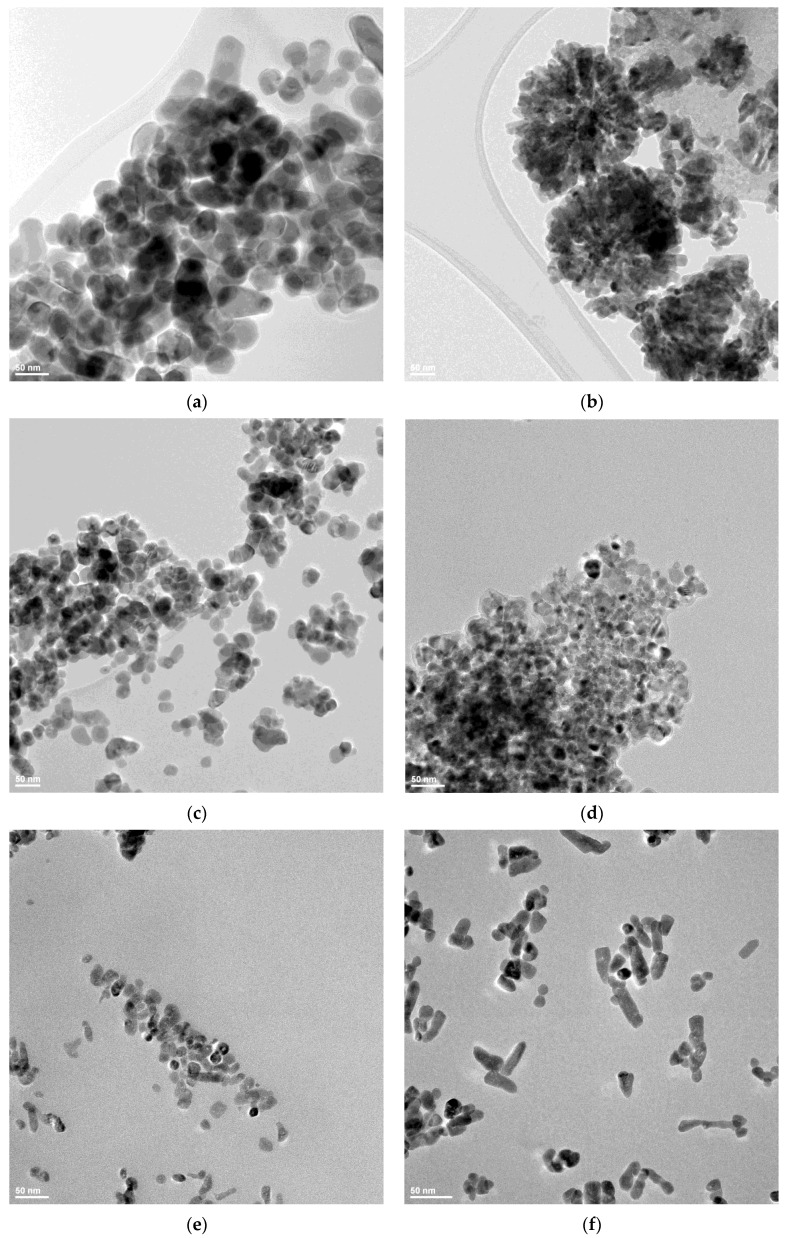
The transmission electron micrographs for: (**a**) ZnO_C1; (**b**) ZnO_C2; (**c**) ZnO_C3; (**d**) ZnO_C4; (**e**) ZnO_C5; (**f**) ZnO_C6 samples.

**Figure 6 pharmaceutics-14-02842-f006:**
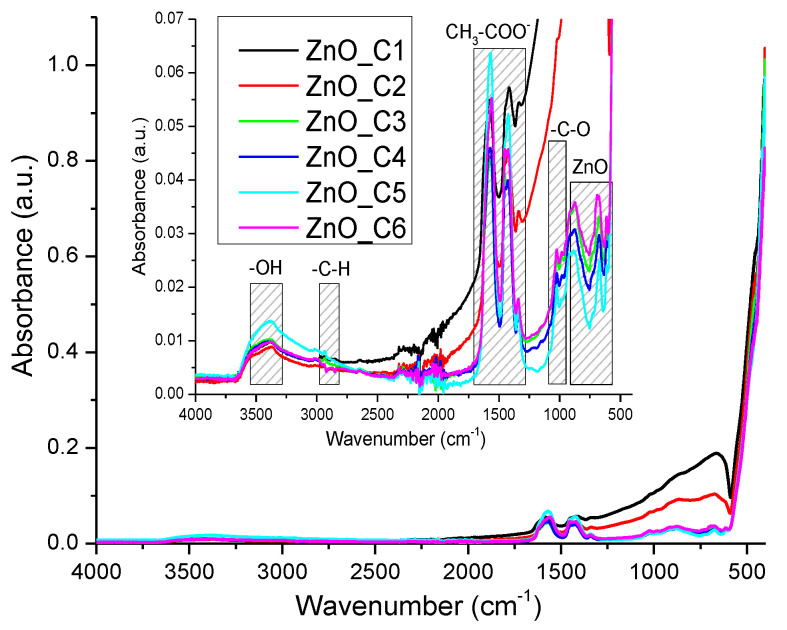
The FTIR spectra for the nanopowders obtained in primary alcohols: ZnO_C1 in methanol; ZnO_C2 in ethanol; ZnO_C3 in 1-propanol; ZnO_C4 in 1-butanol; ZnO_C5 in 1-pentanol; ZnO_C6 in 1-hexanol. Inset presents the detailed spectra between 500–4000 cm^−1^.

**Figure 7 pharmaceutics-14-02842-f007:**
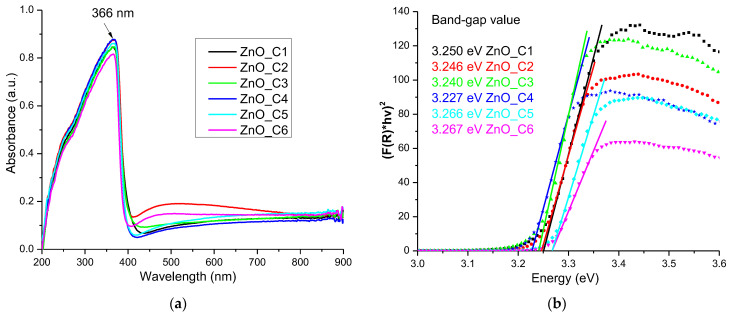
The UV-Vis spectra for the nanopowders obtained in primary alcohols: ZnO_C1 in methanol; ZnO_C2 in ethanol; ZnO_C3 in 1-propanol; ZnO_C4 in 1-butanol; ZnO_C5 in 1-pentanol; ZnO_C6 in 1-hexanol (**a**); the determination of the band-gap values (**b**).

**Figure 8 pharmaceutics-14-02842-f008:**
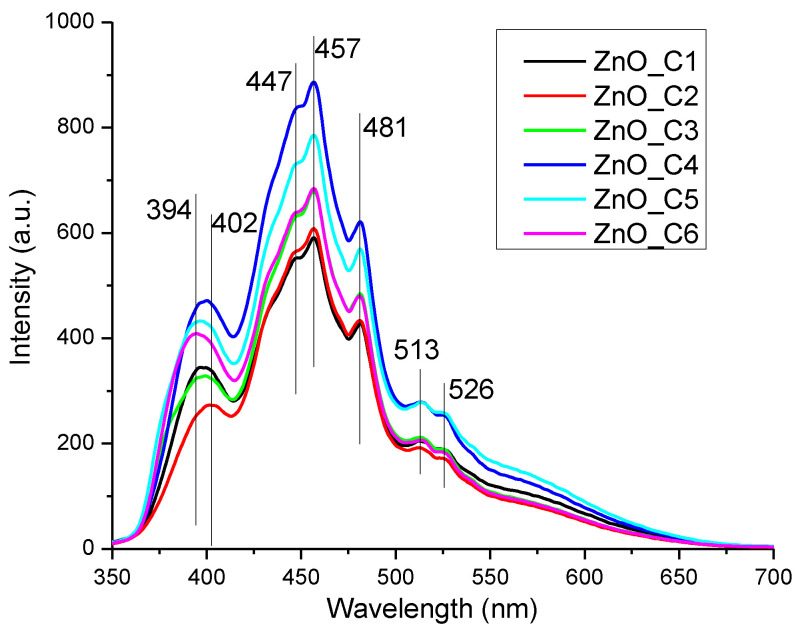
The photoluminescence spectra for the nanopowders obtained in primary alcohols: ZnO_C1 in methanol; ZnO_C2 in ethanol; ZnO_C3 in 1-propanol; ZnO_C4 in 1-butanol; ZnO_C5 in 1-pentanol; ZnO_C6 in 1-hexanol.

**Figure 9 pharmaceutics-14-02842-f009:**
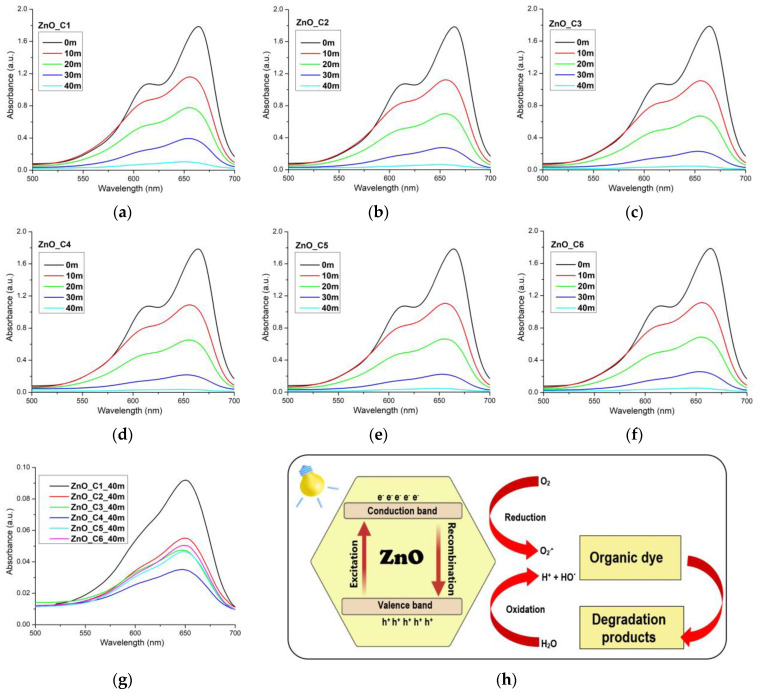
Photocatalytic activity against methylene blue (MB) for: (**a**) ZnO_C1; (**b**) ZnO_C2; (**c**) ZnO_C3; (**d**) ZnO_C4; (**e**) ZnO_C5; (**f**) ZnO_C6; (**g**) detail for all samples after 40 min irradiation; (**h**) the proposed mechanism for photo-degradation of the dye.

**Figure 10 pharmaceutics-14-02842-f010:**
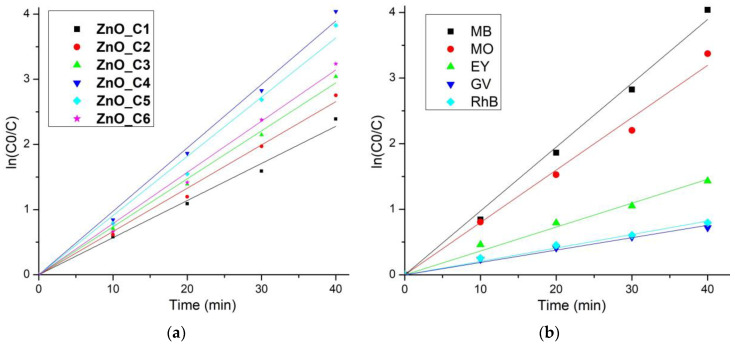
Determination of *k*_app_ from the slope of ln(C_0_/C) vs. time for: (**a**) ZnO_C1 to ZnO-C6 samples against MB solution; (**b**) ZnO_C4 sample against MB, MO, EY, GV and RhB solutions.

**Figure 11 pharmaceutics-14-02842-f011:**
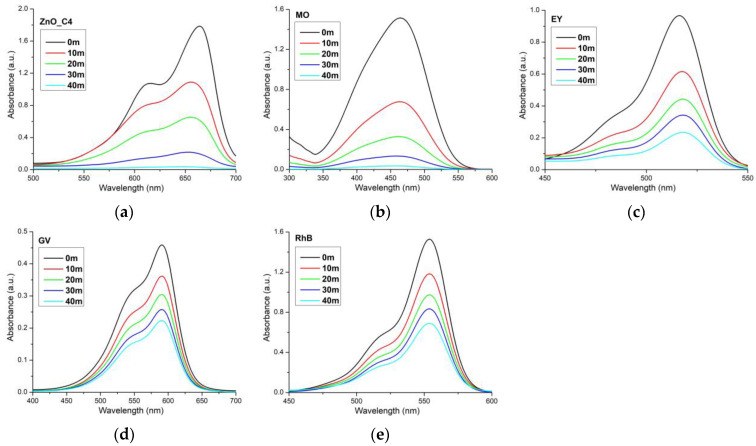
Photocatalytic activity of ZnO_C4 sample against methylene blue (**a**); methyl orange (**b**); eosin Y (**c**); gentian violet (**d**); and rhodamine B (**e**).

**Figure 12 pharmaceutics-14-02842-f012:**
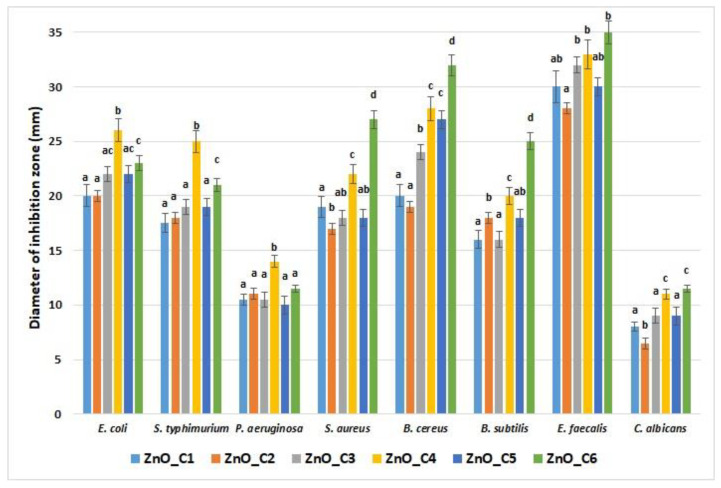
Antimicrobial activity for the nanopowders obtained in primary alcohols: ZnO_C1 in methanol; ZnO_C2 in ethanol; ZnO_C3 in 1-propanol; ZnO_C4 in 1-butanol; ZnO_C5 in 1-pentanol; ZnO_C6 in 1-hexanol. Different small letters indicate statistically significant differences between samples (*p* < 0.05).

**Figure 13 pharmaceutics-14-02842-f013:**
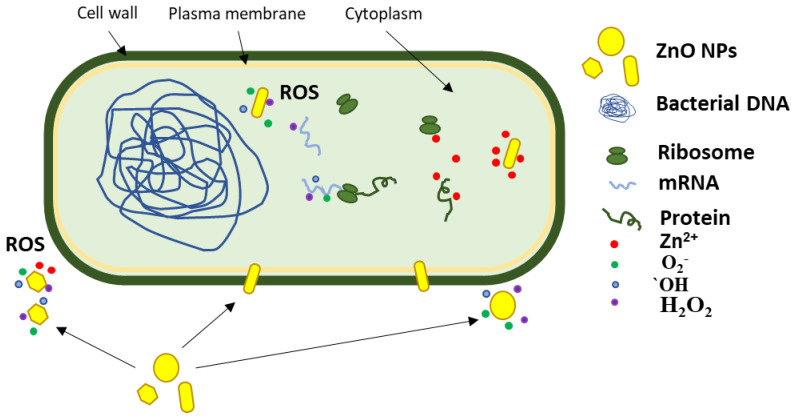
Antibacterial activity mechanisms proposed for the tested ZnO nanoparticles. Generation of reactive oxygen species near bacterial wall and inside the microbial cell is the proposed pathway for small nanoparticles. Mechanical puncture of the plasma membrane, internalization, and disruption of cellular pathways by zinc ions is the proposed mechanism for ZnO nanorods.

**Table 1 pharmaceutics-14-02842-t001:** Sample codification function of the alcohols used and the calculated yield.

Sample Label	Alcohol Used for Synthesis	Note/Yield
ZnO_C1	Methanol	ZnO/97%
ZnO_C2	Ethanol	ZnO/95%
ZnO_C3	1-Propanol	ZnO/96%
ZnO_C4	1-Butanol	ZnO/97%
ZnO_C5	1-Pentanol	ZnO/95%
ZnO_C6	1-Hexanol	ZnO/97%
ZnO_sC3	2-Propanol	White powder/151% *
ZnO_sC4	2-Butanol	White powder/125% *
ZnO_tC4	Tert-butanol	White powder/132% *

* calculated yield assuming theoretical quantity of ZnO (1.845 g).

**Table 2 pharmaceutics-14-02842-t002:** Residual mass at 900 °C for the nanopowders obtained in primary alcohols (ZnO_C1 in methanol; ZnO_C2 in ethanol; ZnO_C3 in 1-propanol; ZnO_C4 in 1-butanol; ZnO_C5 in 1-pentanol; ZnO_C6 in 1-hexanol), in secondary alcohols (ZnO_sC3 in 2-propanol; ZnO_sC4 in 2-butanol) and in a tertiary alcohol (ZnO_tC4 in tert-butanol).

Sample Label	ZnO_C1	ZnO_C2	ZnO_C3	ZnO_C4	ZnO_C5	ZnO_C6	ZnO_sC3	ZnO_sC4	ZnO_tC4
**Residual mass**	93.99%	95.97%	97.66%	98.46%	98.15%	96.75%	44.32%	75.75%	62.43%

**Table 3 pharmaceutics-14-02842-t003:** Lattice parameters for obtained ZnO nanopowders.

Sample Label	ZnO_C1	ZnO_C2	ZnO_C3	ZnO_C4	ZnO_C5	ZnO_C6
**Unit cell**						
***a* = *b* [Å]**	3.25069	3.25056	3.25064	3.25079	3.25099	3.25059
***c* [Å]**	5.20742	5.20716	5.20716	5.20662	5.20728	5.20697
***V* [Å^3^]**	47.65448	47.64824	47.65075	47.65021	47.66206	47.64759
***c*/*a***	1.60194	1.60193	1.60189	1.60165	1.60175	1.60185
**Microstrain (%)**	0.19 ± 0.06	0.25 ± 0.07	0.21 ± 0.06	0.26 ± 0.11	0.21 ± 0.05	0.16 ± 0.04
**Average crystallite size**	54.13 ± 5.79	45.55 ± 2.92	42.59 ± 2.19	34.56 ± 4.98	35.51 ± 1.97	42.05 ± 2.59
**Dislocation density (δ) × 10^−4^**	3.41	4.82	5.51	8.37	7.93	5.66

**Table 4 pharmaceutics-14-02842-t004:** The calculated band-gap energies for the nanopowders obtained in primary alcohols: ZnO_C1 in methanol; ZnO_C2 in ethanol; ZnO_C3 in 1-propanol; ZnO_C4 in 1-butanol; ZnO_C5 in 1-pentanol; ZnO_C6 in 1-hexanol.

Sample Label	ZnO_C1	ZnO_C2	ZnO_C3	ZnO_C4	ZnO_C5	ZnO_C6
**Band-gap value (eV)**	3.250	3.246	3.240	3.227	3.266	3.267

**Table 5 pharmaceutics-14-02842-t005:** The determined values for degradation efficiency, *k*_app_, and R^2^ for ZnO_C1–ZnO_C6 samples against MB solution.

Sample Label	ZnO_C1	ZnO_C2	ZnO_C3	ZnO_C4	ZnO_C5	ZnO_C6
**Degradation efficiency**	94.97%	96.83%	97.85%	98.24%	97.82%	97.36%
***k*_app_ × 10^3^ (min^−1^)**	56.96 ± 1.56	66.43 ± 1.57	73.63 ± 1.31	97.35 ± 2.13	90.84 ± 3.25	78.58 ± 2.03
**R^2^**	0.99624	0.99723	0.99842	0.99761	0.99364	0.99667

**Table 6 pharmaceutics-14-02842-t006:** The determined values for degradation efficiency, *k*_app_, and R^2^ for ZnO_C4 sample against methylene blue (MB), methyl orange (MO), eosin Y (EY), gentian violet (GV) and rhodamine B (RhB) solutions.

Sample Label ZnO_C4	MB	MO	EY	GV	RhB
**Degradation efficiency**	98.24%	97.71%	76.14%	51.34%	54.82%
***k*_app_ × 10^3^ (min^−1^)**	97.35 ± 2.13	79.88 ± 2.47	36.45 ± 1.14	18.91 ± 0.63	20.52 ± 0.65
**R^2^**	0.99761	0.99522	0.99509	0.99446	0.99501

## Data Availability

Not applicable.

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
