# Peer review of "Influence of the Alcohols on the ZnO Synthesis and Its Properties: The Photocatalytic and Antimicrobial Activities"

_pharmaceutics, 2022, doi:10.3390/pharmaceutics14122842_

Round 1

Reviewer 1 Report

Although an important work has been down it does not respond to the microbiological standards and the points I have raised have to be answered.

Although the idea was a good one, and the micoorganisms used are reference strains, I have some critics to the method of measuring microbial inhibition by ZnO nanoparticles

- what was the volume deposited on agar?

- what was the nanoparticle concentrations used ?

- is a possible aggregation of particles taken into account as it may influence the diffusion in agar and thus the results?

- absence of reference and control

The method used is not at all the one use for minimum inhibitory concentration determination as paper disks are deposited on agar, plates surface has to be dry, the solution to be tested is deposited on the disks.

Results are merely qualitative and comparison between the formulations may be disputable.

Author Response

Reviewer 1

Although an important work has been down it does not respond to the microbiological standards and the points I have raised have to be answered.

Answer: 

Following the helpful advices received  we further improved the manuscript and we hope that the esteem reviewer will find it suitable for publishing. We are thankful for the time and effort spent to indicate the areas that needed to be improved.

Point 1:

Although the idea was a good one, and the micoorganisms used are reference strains, I have some critics to the method of measuring microbial inhibition by ZnO nanoparticles

- what was the volume deposited on agar?

- what was the nanoparticle concentrations used ?

Response 1:

We are deeply grateful to the esteem reviewer for pointing out this weakness. We have added the missing information:

“Petri dishes containing nutritive agar were swab inoculate with a standardized inoculum, prepared as bacterial suspensions of 1.5 × 108 CFU/mL (0.5 McFarland) in sterile saline solution (0.9% NaCl). From a previously obtained ZnO suspension (10 mg/mL), drops with a volume of 10 μL were placed on the inoculated Petri dishes and these were incubated for 24 h at 37 °C. The diameter of zone of inhibition of growth developed around each drop was measured (in mm), after incubation.”

Point 2:

- is a possible aggregation of particles taken into account as it may influence the diffusion in agar and thus the results?

Response 2:

As the ZnO is insoluble, we have used a suspension of the nanoparticles. The suspensions were obtained after 30 minutes of ultra-sonication, a step done in order to ensure that nanoparticles are breaking free from all previously agglomerations. Some uncertainty exists when compared with a soluble substance, but this is an additional reason why experiments were made in triplicates.

Point 3:

- absence of reference and control

Response 3:

We have used same volume (10 mL) drops of d.i. water as control.

Point 4:

The method used is not at all the one use for minimum inhibitory concentration determination as paper disks are deposited on agar, plates surface has to be dry, the solution to be tested is deposited on the disks.

Results are merely qualitative and comparison between the formulations may be disputable.

Response 4:

We are grateful to the esteem reviewer for pointing out this weakness. We have enriched the section 3.6 “Antimicrobial assay” and we have discussed comparatively with other reported literature results.

“The literature reports multiple pathways for the antimicrobial activity of the ZnO, as indicated in Figure 13. Production of ROS can take place at the interface between nanoparticles and cellular membrane, after nanoparticles adhesion to the cellular wall [128]. In case of the large particles, clusters and in general nanoparticles that do not come in con-tact with the bacterial membrane, the production of ROS remains the main bactericidal mechanism, as the oxidative stress generated by the accumulation of the ROS can still damage the membrane. Of course, the smaller nanoparticles, with their larger surface generally produce more ROS and therefore exhibit a larger antimicrobial activity [129]. As previously reported smaller nanoparticles tend to cluster near bacterial cell and this also leads to higher bactericidal activity, due to ROS being produced next to the cell wall. However, nanoparticles can penetrate cell membranes and become internalized [130]. ROS production can be resumed inside the microbial cell with more devastating effects. Therefore, the capacity to penetrate the bacterial membrane promotes the nanoparticle to a higher efficiency state, where ROS are delivered inside the cell. Adding to this oxidative stress, the mechanical damage of the membrane, by puncturing or rupture leads to leakage of cytoplasm components like RNA, proteins and K+ and quick death of the cell [128].

X

Figure 13. Antibacterial activity mechanisms proposed for ZnO nanoparticles. Generation of reactive oxygen species near bacterial wall is the proposed pathway for small nanoparticles. Mechanical puncture of membrane, internalization and disruption of mechanisms by zinc ions is the proposed pathway for ZnO nanorods.

Lastly, once inside the microbial cell, the zinc ions can be released from the nanoparticles’ surface, and contribute to the cell death by cytotoxicity, by binding essential components and nutrients of the cell [128]. The real antimicrobial mechanism elucidation re-quires further investigations, but the obtained results permits us to hypothesize that both size and morphology are playing important roles. In a recent study [116], authors found that 6 nm spherical ZnO nanoparticles are more potent than ZnO nanorods with a length of 55 nm, against E.coli and S. aureus. In addition, the group reported that the ZnO nanorods seem to down regulate the expression of Ag43 surface protein, responsible of bacterial adherence and of biofilm formation, which adds to the possible antimicrobial pathways of ZnO.”

Reviewer 2 Report

The manuscript entitled "Influence of the alcohols on the ZnO synthesis and its properties. The photocatalytic and antimicrobial activities" is well-written and very interesting. In this study, the authors reported the influence of the alcohol type on the ZnO synthesis by forced solvolysis. For difference, from the primary alcohols are suitable for obtaining ZnO nanoparticles, from spherical to rod-like shape, the syntheses in secondary and tertiary alcohols were not successful in yielding pure ZnO. Also, the authors showed that the antimicrobial assay indicates that ZnO samples obtained in 1-butanol and 1-hexanol have the best activities. The sample first has the smallest nanoparticles, while the former has rod-shaped particles. This indicates that the shape of the nanoparticles plays an important role in defining the antimicrobial activity of ZnO, most probably due to the mechanical damage and rupture of the cellular membrane. Based on these studies, the authors conclude that the obtained ZnO nanoparticles present a strong antimicrobial activity on a broad range of microorganisms, both bacterial and fungal strains, allowing further use in medical applications. The results of this study are very clearly shown in the Figures.

Author Response

Reviewer 2

The manuscript entitled "Influence of the alcohols on the ZnO synthesis and its properties. The photocatalytic and antimicrobial activities" is well-written and very interesting. In this study, the authors reported the influence of the alcohol type on the ZnO synthesis by forced solvolysis. For difference, from the primary alcohols are suitable for obtaining ZnO nanoparticles, from spherical to rod-like shape, the syntheses in secondary and tertiary alcohols were not successful in yielding pure ZnO. Also, the authors showed that the antimicrobial assay indicates that ZnO samples obtained in 1-butanol and 1-hexanol have the best activities. The sample first has the smallest nanoparticles, while the former has rod-shaped particles. This indicates that the shape of the nanoparticles plays an important role in defining the antimicrobial activity of ZnO, most probably due to the mechanical damage and rupture of the cellular membrane. Based on these studies, the authors conclude that the obtained ZnO nanoparticles present a strong antimicrobial activity on a broad range of microorganisms, both bacterial and fungal strains, allowing further use in medical applications. The results of this study are very clearly shown in the Figures.

Response:

We are grateful to the esteem reviewer for the appreciation words. Following the helpful advices received we further improved the manuscript and we hope that the esteem reviewer will find it suitable for publishing.

Reviewer 3 Report

The paper title "Influence of the alcohols on the ZnO synthesis and its properties. The photocatalytic and antimicrobial activities." reflects exactly what it is about. The literature was checked by me in the Web of Science database in terms of innovation and originality. The synthesis of zinc oxide (and description of its properties) from zinc acetate in ethyl, n-propyl, n-butyl, n-pentyl, and hexyl alcohols was previously investigated ( Parawee et al. Preparation of ZnO nanorod by solvothermal reaction of zinc acetate in various alcohols. Ceramics International 34 (2008) 57–62 and Šarić et al. Alcoholic Solvent Influence on ZnO Synthesis: A Joint Experimental and Theoretical Study J. Phys. Chem. C 2019, 123, 29394-29407). The above publications must be discussed in the final version of the paper. The only difference is that the authors of the reviewed publication did it at a lower temperature of 80 oC, extending the reaction time. In addition, they tested the effect of branched alcohols on the synthesis, which was not investigated previously. The method of testing and describing the properties of zinc oxide (ZnO) should be considered as original, accurate and correct. The authors checked whether there is a correlation between the production of reactive oxygen species and the antimicrobial properties of ZnO obtained in different ways against 8 bacterial strains. The antibacterial properties of ZnO are very well described in the literature, but not in the way presented by the authors as a whole. In conclusion, they say nothing about such a relationship and only state that "This indicates that the shape of the nanoparticles plays an important role in defining the antimicrobial activity for ZnO, most probably due to the mechanical damage induced stress generated by the puncture and rupture of the cellular membrane. Gram-negative strains proved to be more susceptible to the size of the ZnO nanoparticles, while the Gram-positive strains were susceptible to both, smaller size and rod-shape nanoparticles, with a higher influence from the morphology." Based on the results obtained, such a conclusion seems to be possible, but it requires further research on more data and confirmation by other experiments.

The method of citation is not unified.

Author Response

Reviewer 3

The paper title "Influence of the alcohols on the ZnO synthesis and its properties. The photocatalytic and antimicrobial activities." reflects exactly what it is about. The literature was checked by me in the Web of Science database in terms of innovation and originality. The synthesis of zinc oxide (and description of its properties) from zinc acetate in ethyl, n-propyl, n-butyl, n-pentyl, and hexyl alcohols was previously investigated (Parawee et al. Preparation of ZnO nanorod by solvothermal reaction of zinc acetate in various alcohols. Ceramics International 34 (2008) 57–62 and Šarić et al. Alcoholic Solvent Influence on ZnO Synthesis: A Joint Experimental and Theoretical Study J. Phys. Chem. C 2019, 123, 29394-29407). The above publications must be discussed in the final version of the paper. The only difference is that the authors of the reviewed publication did it at a lower temperature of 80 oC, extending the reaction time. In addition, they tested the effect of branched alcohols on the synthesis, which was not investigated previously. The method of testing and describing the properties of zinc oxide (ZnO) should be considered as original, accurate and correct.

Response 1:

We are thankful to the esteem reviewer for indicating these two articles. We have introduce them in our discussion, both in the introduction and at relevant places in XRD and TEM sections.

In both papers, the authors have used autoclave, high temperatures and short reaction time (2-4h). In the study of Tonto P. et al. the XRD analysis indicates that the samples are not pure, low temperatures yielding a heavy unpurified sample with zinc acetate, while for n-butanol even at higher temperature the XRD indicates a large split of all peaks. Nevertheless, one key found was that by increasing the aliphatic chain of the alcohols, the morphology of the particles is changing to longer nanorods, even micron size in 1-decanol. In the second paper, where an additional theoretical study is performed, authors indicate the 1-butanol as the solvent for the smallest nanoparticles to be obtained.

Both studies are valuable addition to compare with and we acknowledged them in the current paper. As a novelty, the current study has the whole range from C1 to C6 primary alcohols and some secondary and tertiary unsuccessful syntheses. Beside the additional characterization techniques, in this study, a full photocatalytic test and antimicrobial assay was performed for the obtained ZnO samples.

The newly introduced information is:

“Forced solvolysis in alcohols without NaOH addition, is seldom reported, and studies still need to be carried out. In one such study Tonto P. et al. have used primary alcohols with 4, 6, 8 and 10 carbon atoms, in an autoclave, at high temperatures of 250-300 oC for 2 h, to obtain nanorods of ZnO. The XRD analysis indicates that at lower temperatures the final product is impurified with zinc acetate. Even at high temperatures, the XRD for the sample obtained in n-butanol presents a large split of the peaks. In a more recent study, Saric A. et al. used primary alcohols with 2, 3, 4, 5 and 8 carbon atoms, also in an autoclave at 170oC for 4 h, to obtain spherical and rod-like nanoparticles. The theoretical modeling of the process indicated the 1-butanol as the solvent in which the smallest crystallite is obtained. Nevertheless, none of these studies was followed up by investigation of the photocatalytic or antibacterial activities.

....................................

In a previous theoretical study [31] the authors have indicated that smallest nanoparticles will be obtained in 1-butanol (when using primary alcohols with 2 to 5 carbon atoms). In addition, the variation of the calculated nanoparticles size is identical with the one we report from our syntheses (DC2 > DC3 > DC5 > DC4).

....................................

The longer aliphatic chains block the increase of the grains on certain directions, allowing the obtaining of rod-shape nanoparticles. A similar hypothesis was made in [30] where authors obtained even micron size rods in 1-decanol.”

Point 2:

The authors checked whether there is a correlation between the production of reactive oxygen species and the antimicrobial properties of ZnO obtained in different ways against 8 bacterial strains. The antibacterial properties of ZnO are very well described in the literature, but not in the way presented by the authors as a whole. In conclusion, they say nothing about such a relationship and only state that "This indicates that the shape of the nanoparticles plays an important role in defining the antimicrobial activity for ZnO, most probably due to the mechanical damage induced stress generated by the puncture and rupture of the cellular membrane. Gram-negative strains proved to be more susceptible to the size of the ZnO nanoparticles, while the Gram-positive strains were susceptible to both, smaller size and rod-shape nanoparticles, with a higher influence from the morphology." Based on the results obtained, such a conclusion seems to be possible, but it requires further research on more data and confirmation by other experiments.

Response 2:

We are grateful to the esteem reviewer for pointing out this weakness. We have enriched the section 3.6 “Antimicrobial assay” and we have discussed comparatively with other reported literature results.

“The literature reports multiple pathways for the antimicrobial activity of the ZnO, as indicated in Figure 13. Production of ROS can take place at the interface between nanoparticles and cellular membrane, after nanoparticles adhesion to the cellular wall [128]. In case of the large particles, clusters and in general nanoparticles that do not come in con-tact with the bacterial membrane, the production of ROS remains the main bactericidal mechanism, as the oxidative stress generated by the accumulation of the ROS can still damage the membrane. Of course, the smaller nanoparticles, with their larger surface generally produce more ROS and therefore exhibit a larger antimicrobial activity [129]. As previously reported smaller nanoparticles tend to cluster near bacterial cell and this also leads to higher bactericidal activity, due to ROS being produced next to the cell wall. However, nanoparticles can penetrate cell membranes and become internalized [130]. ROS production can be resumed inside the microbial cell with more devastating effects. Therefore, the capacity to penetrate the bacterial membrane promotes the nanoparticle to a higher efficiency state, where ROS are delivered inside the cell. Adding to this oxidative stress, the mechanical damage of the membrane, by puncturing or rupture leads to leakage of cytoplasm components like RNA, proteins and K+ and quick death of the cell [128].

X

Figure 13. Antibacterial activity mechanisms proposed for ZnO nanoparticles. Generation of reactive oxygen species near bacterial wall is the proposed pathway for small nanoparticles. Mechanical puncture of membrane, internalization and disruption of mechanisms by zinc ions is the proposed pathway for ZnO nanorods.

Lastly, once inside the microbial cell, the zinc ions can be released from the nanoparticles’ surface, and contribute to the cell death by cytotoxicity, by binding essential components and nutrients of the cell [128]. The real antimicrobial mechanism elucidation re-quires further investigations, but the obtained results permits us to hypothesize that both size and morphology are playing important roles. In a recent study [116], authors found that 6 nm spherical ZnO nanoparticles are more potent than ZnO nanorods with a length of 55 nm, against E.coli and S. aureus. In addition, the group reported that the ZnO nanorods seem to down regulate the expression of Ag43 surface protein, responsible of bacterial adherence and of biofilm formation, which adds to the possible antimicrobial pathways of ZnO.”

Point 3:

The method of citation is not unified.

Response 3:

We are thankful for pointing out this weakness. We added article number/pages as needed in references.

Reviewer 4 Report

The reported work "Influence of the alcohols on the ZnO synthesis and its properties. The photocatalytic and antimicrobial activities" is interesting and can be considered for publishing, after the following improvements/revision,

1- The authors use Perkin Elmer P55 (Perkin Elmer, Waltham, MA, USA) spectrometer with Xe lamp with an excitation wavelength of 320 nm Considering the experimental setup and description provided in fig.8, please clarify the followings,

Emission spectra are recorded between 350-700 nm and there is very little difference between excitation and emission wavelength. No filter was used to avoid the backscattering of Xe lamp. It might be possible that the authors are recording emission from Xe lamp as well.

Line No. 309, please explain what is exciton recombination.

Line No. 313 – 319, please provide the experimental evidence supporting oxygen vacancies (VO), zinc vacancies (VZn), oxygen anti-sites (OZn), oxygen interstitials (Oi) or zinc interstitials (Zni).

2- Thermal analysis was performed under a flow of 50 mL∙min−1 of dried air. Please explain the percentage of oxygen and nitrogen in the dry air. Or is it only nitrogen? Also, the description of/from Fig.  1-3 and table 2 is insufficient.

TG analysis of ZnO C6 in the range 300-400 experience more intense exothermal reactions than other samples which show similar or decreasing tendencies. What is the explanation of the obtained results?

The majority of the samples in TG analysis in the range 400-900 show negative DSC values connected with endothermal reactions. The observed phenomena should be described.

Table 2 should also include the residual mass of nanopowders from secondary and tertiary alcohols for comparison with primary alcohols synthesis

I suggest taking 1st derivative (DTG) of the TG curve and explaining the weight loss in steps. This is important for the readers to understand the relative thermal decomposition with an increase in temperature because of the different elements in the sample. Authors can take help and enrich their discussion from doi.org/10.1016/j.jclepro.2022.133853.

3- In the TEM images size labels are missing for the chosen nanoparticles/nanorods

4- The obtained results (photocatalysis) should be compared with other types of photoactive nanomaterials from literature, such as silver (doi.org/10.1371/journal.pone.0274753) and titania (doi.org/10.1016/j.jhazmat.2021.126958)

Author Response

Reviewer 4

The reported work "Influence of the alcohols on the ZnO synthesis and its properties. The photocatalytic and antimicrobial activities" is interesting and can be considered for publishing, after the following improvements/revision,

Answer:

Following the helpful advices received, we further improved the manuscript and we hope that the esteem reviewer will find it suitable for publishing. We are thankful for the time and effort spent to indicate the areas that needed to be improved.

Point 1:

1- The authors use Perkin Elmer P55 (Perkin Elmer, Waltham, MA, USA) spectrometer with Xe lamp with an excitation wavelength of 320 nm Considering the experimental setup and description provided in fig.8, please clarify the followings,

Emission spectra are recorded between 350-700 nm and there is very little difference between excitation and emission wavelength. No filter was used to avoid the backscattering of Xe lamp. It might be possible that the authors are recording emission from Xe lamp as well.

Response 1:

We are very grateful to the esteem reviewer for pointing out this mistake. A spectrum recorded with no cut-off filter, only with the 1% attenuator, would have a U shape on sides, at 350 nm and at 610 nm (~30 nm higher than excitation wavelength used and ~30 nm lower than double of the excitation wavelength).

Without cut-off filter, it is not possible to measure around 2x excitation wavelength, namely 640 nm. Obviously, this is not the case here. We have corrected the information, both in methods section and in discussion:

“A Perkin Elmer LS55 (Perkin Elmer, Waltham, MA, USA) fluorimeter was used to record the photoluminescence spectra (PL). A Xe lamp was used as an excitation source at ambient temperature. The excitation wavelength was 320 nm. The emission spectra were recorded in the domain 350–700 nm, with a scan speed of 200 nm min−1 and a 350 nm cut-off filter.”

Point 2:

Line No. 309, please explain what is exciton recombination.

Response 2:

Excitons are much alike hydrogen atoms except that a positive hole and not a proton is the partner of the electron. They are thus electron–hole pairs bound by electrostatic interaction. They can form in any semiconductor.

We have added the information:

"......by the free exciton recombination (recombination of the electron and hole pair that was previously generated by the absorption of a photon)."

Point 3:

Line No. 313 – 319, please provide the experimental evidence supporting oxygen vacancies (VO), zinc vacancies (VZn), oxygen anti-sites (OZn), oxygen interstitials (Oi) or zinc interstitials (Zni).

Response 3:

We discussed more the types of vacancies that occurs in ZnO, and indicated literature reports where the mechanisms, defects and transitions are presented. The fluorescence spectrum per se is the main indicator that these defects do exist in a sample, otherwise only the UV-emission band would appear.

Point 4:

2- Thermal analysis was performed under a flow of 50 mL∙min−1 of dried air. Please explain the percentage of oxygen and nitrogen in the dry air. Or is it only nitrogen? Also, the description of/from Fig.  1-3 and table 2 is insufficient.

Response 4:

We have added the requested information. The equipment uses hydrocarbons free, dry air from Linde. As such, it contains 20% O2 and 80% N2, and the dynamic atmosphere for this analysis must be considered oxidative.

"The samples (~20 mg of dry powder) were placed in an open Al2O3 crucible and heated up with a 10 oC∙min−1 rate until 900 °C, in dynamic atmosphere (flow of 50 mL∙min−1 of dried air – 20% O2 and 80% N2)."

The description of Figures 1-3 and Table 2 were further detailed.

Point  5:

TG analysis of ZnO C6 in the range 300-400 experience more intense exothermal reactions than other samples which show similar or decreasing tendencies. What is the explanation of the obtained results?

Response 5:

 After 300oC the combustion of organic impurities and evolved gaseous products is giving rise to the exothermic peak. The peak intensities are partially related to the speed of the oxidations processes.

Point 6:

The majority of the samples in TG analysis in the range 400-900 show negative DSC values connected with endothermal reactions. The observed phenomena should be described.

Response 6:

We have added the following possible explication as founded in the article indicated by the esteem reviewer.

"In all samples, an endo-thermic effect after 400 oC is probably related to the elimination of surface –OH moieties, densification and crystallization processes [87]."

Point 7:

Table 2 should also include the residual mass of nanopowders from secondary and tertiary alcohols for comparison with primary alcohols synthesis

Response 7:

We have included the residual mass from secondary and tertiary alcohols in the Table 2.

Point 8:

I suggest taking 1st derivative (DTG) of the TG curve and explaining the weight loss in steps. This is important for the readers to understand the relative thermal decomposition with an increase in temperature because of the different elements in the sample.

Authors can take help and enrich their discussion from doi.org/10.1016/j.jclepro.2022.133853.

Response 8:

We are thankful for the suggestion. We have included all six thermal analyses with details in the supplementary file. The interpretation part for the TG/DSC analysis was enriched with the excellent article suggested by the esteem reviewer.

"Results for individual thermal analyses made for samples obtained in primary alcohols are given in Supplementary materials, Figures S1-S6, and the principal data (mass loss and effect temperatures) are listed in Table S1.

..................................

After 240 oC the acetate impurities are starting to decompose, releasing acetic acid and acetone, this process being accompanied by an endothermic effect on the DSC curve in the interval 260-290 oC [86]. After 300 oC the remaining organic impurities will be oxidized, together with the evolved products from decompositions as indicated by the exothermic peaks on the DSC curve in the 330-355 oC interval. The exothermic peaks from this interval are presenting a shoulder, which can be attributed to the oxidation of residual carbonaceous mass from previous incomplete oxidation processes [87]. In all samples, an endo-thermic effect after 400 oC is probably related to the elimination of surface –OH moieties, densification and crystallization processes [87]."

Point 9:

3- In the TEM images size labels are missing for the chosen nanoparticles/nanorods

Response 9:

Indeed, the size bars in TEM images are at 50 nm, with white font in left bottom corner and not very visible.

Point 10:

4- The obtained results (photocatalysis) should be compared with other types of photoactive nanomaterials from literature, such as silver (doi.org/10.1371/journal.pone.0274753) and titania (doi.org/10.1016/j.jhazmat.2021.126958)

Response 10:

We thank esteem reviewer for indicating this valuable studies. We have added comparison with them in relevant places or mention them as ROS generators and therefore as strong antimicrobial materials.

"Many materials generate ROS in the presence of light [37], and ZnO is one with a high capacity of ROS generation. Such materials usually exhibit a strong antimicrobial activity.

.................................................................................................

Worth noted that all three compounds EY, GV and RhB are part of same class of triarylmethane dyes, this common structure bestowing them a higher resistance to pho-to-degradation [115], although some literature reports indicate a high removal rate for titanium doped activated carbon nanostructures against such dyes [116]."

Round 2

Reviewer 1 Report

The revision of the text is acceptable although the picture of bacteria is a little "strange"(looks like a vibrio!) and should be modified. The nano particles are  not at the same scale than the microorganism and the bacterial shape is not an usual one. The "membrane" is not correct as there are two  layers (cell wall, or outer membrane, and cytoplasmic membrane separated by the periplasmic space), thus if nanorods are entering the cell they certainly cannot do this in one step even if the proposal for explaining the higher activity of rods over spheric nanoparticle is acceptable.

I prefer "plasmic membrane" in lieu of "membrane" which is a little bit confusing.

It is not surprising that the smaller nanoparticles have a more potent activity are they are susceptible to cross the cell wall easily and it may be also the case for nanorods, provide their diameter is small enough.

Otherwise the paper is acceptable.

Author Response

Reviewer 1

The revision of the text is acceptable although the picture of bacteria is a little "strange"(looks like a vibrio!) and should be modified. The nano particles are  not at the same scale than the microorganism and the bacterial shape is not an usual one. The "membrane" is not correct as there are two  layers (cell wall, or outer membrane, and cytoplasmic membrane separated by the periplasmic space), thus if nanorods are entering the cell they certainly cannot do this in one step even if the proposal for explaining the higher activity of rods over spheric nanoparticle is acceptable.

I prefer "plasmic membrane" in lieu of "membrane" which is a little bit confusing.

It is not surprising that the smaller nanoparticles have a more potent activity are they are susceptible to cross the cell wall easily and it may be also the case for nanorods, provide their diameter is small enough.

Otherwise the paper is acceptable.

Answer:

Once again, the esteem reviewer is correct about the scale and constitution of the microorganism’s cells.

We have remade figure 13 according to the instructions and enriched it.

Membrane was replaced with plasma membrane.

We are very grateful to the esteem reviewer for his/her appreciation and we are thankful for the time and effort spent to help us improve the manuscript.

Reviewer 4 Report

The authors have made a sincere effort to improve the manuscript. The manuscript now can be considered for publication.

Author Response

Reviewer 4

The authors have made a sincere effort to improve the manuscript. The manuscript now can be considered for publication.

Response

We are very grateful to the esteem reviewer for his/her appreciation and we are thankful for the time and effort spent to help us improve the manuscript.